# The Concurrent Sintering-Crystallization Behavior of Fluoride-Containing Wollastonite Glass-Ceramics

**DOI:** 10.3390/ma14030681

**Published:** 2021-02-02

**Authors:** Chuanhui Li, Peng Li, Jianliang Zhang, Fengjuan Pei, Xingchen Gong, Wei Zhao, Bingji Yan, Hongwei Guo

**Affiliations:** 1School of Metallurgical and Ecological Engineering, University of Science and Technology Beijing, Beijing 100083, China; jgzmhui@foxmail.com (C.L.); jl.zhang@ustb.edu.cn (J.Z.); 2School of Iron and Steel, Soochow University, Suzhou 215021, China; zhaowei0312@suda.edu.cn (W.Z.); bjyan@suda.edu.cn (B.Y.); guohongwei@suda.edu.cn (H.G.); 3School of Materials Science and Engineering, University of Science and Technology Beijing, Beijing 100083, China; pfj529@126.com; 4School of Metallurgical Engineering, Anhui University of Technology, Ma’anshan 243002, China; agdgxc@163.com

**Keywords:** crystallization, fluoride, glass-ceramics, sintering

## Abstract

The fabrication of well densified wollastonite with smooth appearance by direct sintering method is still a challenge due to the competitive behaviors between sintering and crystallization. In this study, the coarser glass frits with a size of 1–4 mm are subjected to heat treatment at different temperatures. An attempt of integrating differential thermal analyzer with a slag melting temperature characteristic tester was exploited to monitor the heat and geometry changes during the heating. The results showed that the addition of CaF_2_ can significantly promote the crystallization of wollastonite at 940 °C, while hindering the sintering ability. At higher temperature, the increase of CaF_2_ acts as flux and favors the formation of eutectics, leading to a decline in the precipitation amount of wollastonite. The predominated liquid sintering brought fast shrinkage. It was found out that high content of CaF_2_ narrows the dense sintering temperature range and results in uneven surfaces. In order to obtain wollastonite glass-ceramics with smooth appearance, the maximum content of CaF_2_ in sintering glass-ceramics should be limited to 2 wt.%.

## 1. Introduction

Wollastonite is a calcium inosilicate mineral (CaSiO_3_), which belongs to the pyroxenoid group of white inorganic materials with a ratio of Si:O = 1:3. The polymeric transition from para-wollastonite (β-CaSiO_3_) to pseudo-wollastonite (α-CaSiO_3_) occurs when temperature is above 1125 °C [1]. Wollastonite glass-ceramics possess outstanding characteristics, i.e., high whiteness, low moisture absorption, low thermal expansion, low shrinkage, and low dielectric constant, which are suitable for wide applications in ceramics, chemicals, electronic devices, dental implant, construction, and polymers [2,3,4,5]. 

The fluorides are often precipitated in CaO-SiO_2_ system owing to the antibacterial effect of F^−^ in biomedicine application [6]. The low amount of fluorides could promote the nucleation and crystallization of wollastonite [7], whereas the high concentration of fluorides leads to the precipitation of fluorosilicates (KMg_3_AlSi_3_O_10_F_2_, KNaCaMg_5_Si_8_O_22_F_2_), fluorapatite (Ca_5_(PO_4_)_3_F), and CaF_2_, depending on its glass composition [4,6,7,8]. The preparation of wollastonite-based glass-ceramics with well densified structure is still a challenge via direct sintering process. Compared with the traditional melting process, direct sintering is capable of initiating crystallization without adding nucleation agents, reducing the processing temperature and energy consumption [8]. The raw materials are melted and quenched into fine powders, then subjected to heat treatment for crystallization at relatively lower temperatures [9]. The crystallization and sintering mechanisms are usually in competition when direct sintering method is adopted for glass-ceramics’ preparation, which should be delicately balanced to produce densified structure with smooth surface. Theoretically, the key when fabricating dense glass-ceramics is to make the glass particles sinter to almost their full density by viscous flow before crystallization. 

To understand what is occurring during the crystallization, one needs to monitor the microstructural change as a function of the heat treatment variables, such as time, temperature, and particle size [10,11]. This study attempted to provide a simple methodology that combines the techniques of differential thermal analyzer (DTA) and a slag melting temperature characteristic tester, with an aim of recording the exothermic reactions and the geometry changes during the heating. The variations in characteristic temperatures correlated to the thermal events of wollastonite-based glass-ceramic with respect to different CaF_2_ additions were analyzed. Hopefully, the essential knowledge gained in this work will be useful for engineering use. 

## 2. Materials and Methods

The raw materials for wollastonite glass-ceramic were reagent-grade chemicals and the batch compositions were adopted from the Japanese company “Nippon Electric Glass”, which is named Neoparies [12]. As listed in Table 1, samples F1–F4 were prepared by introducing fluorine in the form of CaF_2_ to 100 g of basic composition with verifying amounts of 1, 2, 3, 4 g, respectively. Figure 1 shows the process flow diagram for the preparation of wollastonite glass- ceramic and subsequent characterizations. After mixing, the batch mixture was transferred into the platinum crucible and melted at 1500 °C for 2 h in a muffle furnace (GWL-1600, Luoyang Juxing Kiln Cos., Ltd., Luoyang, China) with air atmosphere. The melt was then quickly quenched in distilled water to prevent crystallization. The obtained glass frits were dried at 80 °C and crushed into coarse particles with a particle size of 1–4 mm, then heated at 940, 1000, 1060, 1120, and 1180 °C for 1 h to obtain glass-ceramics.

XRD analysis. The crystalline phases presented in the synthesized glass-ceramic samples were identified by an Ultima IV X-ray diffraction (Rigaku, Tokyo, Japan). The samples were subjected to grinding, then sieved for 5 min (mesh size of 74 μm). The obtained powder (≤74-μm particle size) was scanned between 2θ = 10–70° with a step size of 2θ = 0.02° and a scanning speed of 2°/min. The phase composition of glass-ceramic samples (i.e., the amorphous and the crystalline phases’ amounts) was determined by the combined Rietveld-reference intensity ratio (RIR) methods using α-Al_2_O_3_ as an internal standard [13] and the calculated equation was as follows:
(1)$$X_{c}=\frac{I_{c}}{I_{c}+I_{a}}$$
where *X_c_* is the crystallinity of the test sample and *I_c_* and *I*_a_ are the diffraction intensity of crystalline and amorphous phase in the test sample, respectively.

SEM-EDS analysis. The fractured surfaces of the prepared glass-ceramics’ samples were polished and chemically etched for 3–5 s in 4 vol% hydrofluoric acid solution, followed by microstructure characterization by a field emission scanning electron microscope (SEM; SU5000, Hitachi, Japan). Meanwhile, energy dispersive spectroscopy (Oxford EDS X-MAX 20, Oxfordshire, UK) analysis was adopted to identify specific elements. 

DTA analysis. The thermal stability of glass powders (particle size ≤ 74 μm) was characterized by a differential thermal analyzer (DTA; Labsys Evo Simultaneous Thermal Analysis, SETARAM, Caluire-et-Cuire, France) in argon atmosphere. The sample was subjected to heating from room temperature to 1200 °C at a heating rate of 10 °C/min, and the crystallization peak temperature (T_p_) for each glass sample was measured. 

Raman analysis. Raman spectroscopy (HR800, HORIBA JobinYvon, France) for the parent glass samples was performed to determine the influence of F^−^ ions on the melt structure units.

Sintering analysis. As shown in Figure 2, a slag melting temperature characteristic tester (CQKJ-II, Chongqing University of Science & Technology, Chongqing, China), equipped with an image recording system and electrical furnace, was used to record the geometry changes of the glass samples during the sintering. The glass powder (particle size ≤ 74 μm) was compressed into a cylinder (with a diameter of 3 mm and height of 3 mm) and then heated to 1300 °C in air with a heating rate of 10 °C/min. The sample images were analyzed using computer software to calculate the height H_T_ and the area A_T_ of the outline of the sample. The characteristic temperatures in Table 2 correlated to sintering events were interpreted based on the geometry changes. The initial height H_0_ and area A_0_ were measured at 500 °C and used as a reference. The average value of three replicates of respective samples were collected to reduce measurement error.

## 3. Results

### 3.1. The Wollastonite Formation at Different Temperatures

Figure 3 shows the XRD results of the glass-ceramic samples after heat treatment at 940 °C and 1180 °C for 60 min. Peaks corresponding to para-wollastonite (PDF #10-0489) were identified at 940 °C as the only crystalline phase, along with an amorphous silicate hump located at 20–35°. A phase transition from para-wollastonite to pseudo-wollastonite (PDF #74-0874) was observed when temperature reached 1180 °C. Figure 4 shows the extent of crystallization. After heat treatment at 940 °C, the intensity of para-wollastonite peak was greatly enhanced by CaF_2_ addition. The corresponding crystallinity was increased from 19.71% to 35.18%. 

As shown in Figure 5, Raman spectra between 1200–800 cm^−1^ represent symmetric stretching vibrations of [Si-O] structure units and were deconvoluted to quantitatively estimate the relative fractions of Q^n^ units (Q^n^: referring to symmetric stretching vibrations of units with n numbers of bridging oxygen, Q^0^-[SiO_4_]^4−^, Q^1^-[Si_2_O_7_]^6−^Q^2^-[Si_2_O_6_]^4−^, Q^3^-[Si_2_O_5_]^2−^, Q^4^-SiO_2_) [7,15,16,17,18]. According to Equations (2) and (3), the increment of Ca^2+^ and F^−^ ions led to depolymerization of silicate glass. Due to the similar ion radius, the partial replacement of Si-O-Si bonds by F^−^ will result in the formation of Si-F bonds. As a consequence, [Si_2_O_5_]^2−^ sheets (one non-bridging oxygen atom per silicon) would transform into [SiO_3_]^2−^ units (two non-bridging oxygen atoms per silicon) [17,18]. This was strongly validated by the results of the deconvolution. It is clearly seen from Figure 5F, the fraction of Q^2^ units increased with the addition of CaF_2_ at the expense of Q^3^ units, whereas the fractions of the Q^0^, Q^1^, and Q^4^ units do not show much variation. Moreover, F^−^ ions entered into the silicate glass and facilitated the glass separation. Both effects favored the precipitation of wollastonite. When the heat treatment temperature reached 1000 °C, the crystallinity had the highest value. However, higher CaF_2_ addition did not bring higher crystallinity. On the contrary, it became lesser, and a much steeper decline was observed when temperature reached 1180 °C.
(2)$${\left[{\mathrm{Si}}_{2}O_{5}\right]}^{2-}+2F^{-}\to {\left[{\mathrm{SiO}}_{2}F_{2}\right]}^{2-}+{\left[{\mathrm{SiO}}_{3}\right]}^{2-}$$
(3)$${\left[{\mathrm{SiO}}_{3}\right]}^{2-}+{\mathrm{Ca}}^{2+}\to {\mathrm{CaSiO}}_{3}$$


### 3.2. The Microstructure Characterization of Glass-Ceramics

Figure 6 shows SEM images of the glass-ceramics with different CaF_2_ contents after heat treatment at 1120 °C for 60 min. Dense body was found in all glass-ceramic specimens. White fibrous and dendritic grains with a length of 10 μm were identified as wollastonite phase based on EDS analysis (Table 3) and were the only crystalline phase observed on the etched fracture surface. The high concentration of fluorine element was derived from HF acid etching. It was hard to distinguish whether the fluoride was actually dissolved into the wollastonite structure to form a solid solution. Salman et al. suggested wollastonite could acquire considerable amounts of Na^+^, K^+^, Fe^2+^, and Mg^2+^, as well as F^−^, since the radius of last (1.36 Å) is close to that of O^2−^(1.40 Å) and could replace O^2−^ easily [4]. With the increment of CaF_2_ content, the precipitation amount of wollastonite grains became fewer but in much larger size (see Figure 6d). Due to its bone-breaking properties, the CaF_2_ addition to parent glass can lower the viscosity, with a consequence of the reduction of the melting temperature and inter-particle free surface [19,20,21]. Since the nucleation of wollastonite is governed by surface nucleation by direct sintering method, the number of heterogeneous nucleation sites of wollastonite would decrease. Meanwhile, the mass transfer for wollastonite growth could be enhanced by lower viscosity, thereby forming larger wollastonite grains.

### 3.3. Influence of CaF_2_ Content on Concurrent Sintering and Crystallization Behavior

During the fabrication of glass-ceramics, the controlling of concurrent events, i.e., crystallization and sintering, is essential to ensure the final product quality. For better understanding the relationship between these two events, shrinkage and DTA curves of samples F1–F4 are shown in Figure 7. The correlated characteristic temperatures for glass samples were extracted from the curves. According to Table 4, with increasing CaF_2_ content, the initial sintering temperature T_fs_ decreased from 775 °C to 760 °C, and the temperature of maximum shrinkage T_ms_ decreased from 895 °C to 870 °C. Since diffusion and viscous flow are responsible for sintering, it is clear that the presence of CaF_2_ will greatly enhance the sintering kinetics. In this case, CaF_2_ in glass acted as a powerful network disrupter, weakening the structure of silica network and improving the mass migration [7,8]. At this stage, the sintering necks between adjacent particles were developed and the cavities were filled by viscous flow. 

Based on the DTA curves, an exothermic peak corresponding to crystallization reaction appeared during the sintering process. When CaF_2_ content increased from 1 to 4 wt.%, the initial crystallization temperature T_x_ decreased from 815 °C to 805 °C and the crystallization peak temperature T_p_ decreased from 858 °C to 840 °C. These results suggest that the addition of CaF_2_ could lower the glass stability and promote crystallization, which is in accordance with the XRD results. It is worth noting that the T_x_ of all samples was lower than their T_ms_, indicating the crystallization occurred before they reached their maximum shrinkage values before softening. The ability of crystallization vs. sintering, ΔT_1_ = T_ms_ − T_x,_ was evaluated to justify the interactions. The larger value of 42 °C obtained by F2 sample indicated a larger final density could be achieved with 2 wt.% addition of CaF_2_. After the crystalline phase was precipitated, consequently, the slope of shrinkage curves was dramatically decreased. Figure 7 shows that, when the sintering temperature rose to T_ms_, the height of samples F1–F4 became 73.8% (F1), 73.5% (F2), 75.2% (F3), and 74.7% (F4) of its initial height H_0_. It implied that the crystallization that occurred on the surfaces of the glass particles hindered the sintering process and reduced the sample shrinkage. 

As the temperature further increased, the densification process was suspended, as indicated by a plateau stage, and lasted until the cylinder sample became softened. At this period, the continuous crystallization increased the viscosity of the residual glass and froze the sintering. It might be the potential reason why the samples with higher addition of CaF_2_ had the lower T_ms_. When the temperature approached the onset temperature of softening T_d_, a dramatic decline trend in height was observed. The residual glass, the viscosity of which viscosity was greatly reduced at high temperatures, accelerated the liquid phase sintering of powder compacts. The samples began to soften and shrink faster. Based on Table 4, the softening temperatures T_d and_ T_s_ as well as the hemispheric temperature T_hb_ for samples F1–F4 increased with the addition of CaF_2_. However, the flowing temperature exhibited an opposite trend. The former can be ascribed to the enhanced precipitation of wollastonite phase with high melting point, which led to an increment in T_d,_ T_s_, and T_hb_. The latter was due to the fluxing effect of CaF_2_, which led to degraded refractoriness of glass-ceramics. Deng et al. found out that the addition of a moderate amount of CaF_2_ in CaO-Al_2_O_3_-MgO-SiO_2_ glass led to a decrease in the melting temperature and viscosity [22]. In this study, the dense sintering temperature range was proposed as an important index for evaluating sintering ability of glass-ceramics and was defined as ΔT_2_ = T_hb_ − T_d_. High-quality glass-ceramics with smooth surfaces require a broad, dense sintering temperature range, which is of great benefit to practical production. It is evident from Table 4, increasing CaF_2_ amount decreased ΔT_2_ from 70 °C to 20 °C. Based on the results from Table 4 and Figure 8, the dense sintering temperature ΔT_2_ of glass-ceramics should be greater than 30 °C in order to obtain a smooth surface. Therefore, the content of CaF_2_ for wollastonite glass-ceramics was limited in 2 wt.%.

Based on the above results, three stages of sintering process were identified. At the first stage, sintering necks formed between the adjacent glass particles, and the interparticle contact area increased by neck growth. The viscous flow led to stronger inter-particle bonding and the elimination of pores. The second stage had a large temperature range, from approximately 900 to 1100 °C, where the densification was frozen due to the crystallization. As shown in Figure 8, the rough surface of the glass-ceramic samples was observed. Clearly, compared with sintering, the crystallization was more dominant for samples containing higher CaF_2_ content.

The third stage was characterized by fast shrinkage, which derived from the liquid formation as the temperature approached the softening temperature. At this stage, the liquid sintering became the dominant process. CaF_2_ acted as flux agent to form low-temperature eutectics. An increasing amount of melt would be transferred from the particle surface or bulk into the neck area via surface or volume diffusion and result in enhanced sintering kinetics. In addition, the deformation and rearrangement of the particles favored flattening the sample surfaces. Along with the sintering, as confirmed in XRD analysis, the precipitation amount of wollastonite in equilibrium with the melt decreased. For samples containing lower CaF_2_, most of the sintering necks disappeared, resulting in a flat glass-ceramic appearance while maintaining considerable crystallinity (F1-1120 °C, F2-1180 °C). F3 and F4 samples, owing to the lower flowing temperature, had a low crystallinity and exhibited over-burnt surface. 

It is well known that to obtain well-densified glass-ceramics, crystallization has to be suppressed until most of the shrinkage has occurred. Our work demonstrated a compromised balance between sintering and crystallization could be reached, in order to yield dense glass-ceramics. In this regard, two potential routes were proposed by implementing two stages of heat treatment at the expense of the crystallinity. During that, the sequence of sintering and crystallization can be properly adjusted: (1) The precipitation of crystalline phases took place first, then was followed by liquid sintering at high temperature, or (2) conducting liquid sintering to ensure a flat surface, then subjected to low temperature crystallization.

## 4. Conclusions

To produce well-densified, wollastonite-based glass-ceramics with smooth appearance, it was suggested the content of fluoride should be limited in 2 wt.%. The influences of CaF_2_ on the sintering and crystallization behavior can be drawn as follows:
(1)As the amount of CaF_2_ increased, both the onset crystallization temperature T_x_ and temperature of maximum shrinkage T_ms_ for glass-ceramics decreased. The depolymerization effect of fluoride led to the enhancement in crystallization. The accompanying grain boundary creation and coarsening reduced the shrinkage rate and resulted in uneven appearance.(2)At high temperature, the dominant liquid sintering brought severe shrinkage and a smooth appearance for the product. Accordingly, the precipitation amount of wollastonite decreased with the increment of CaF_2_, due to its characteristic fluxing effect. The dense sintering temperature range (ΔT_2_ = T_hb_ − T_d_) was inversely related to the fluoride content, implying that high CaF_2_ addition was detrimental to final properties of product.


In this regard, the evaluations of sintering and crystallization kinetics will be implemented with mathematical models in future work. More microstructural information, such as grain coarsening and pore development, will be of great use for a better understanding of the concurrent sintering-crystallization behavior.

## Figures and Tables

**Figure 1 materials-14-00681-f001:**
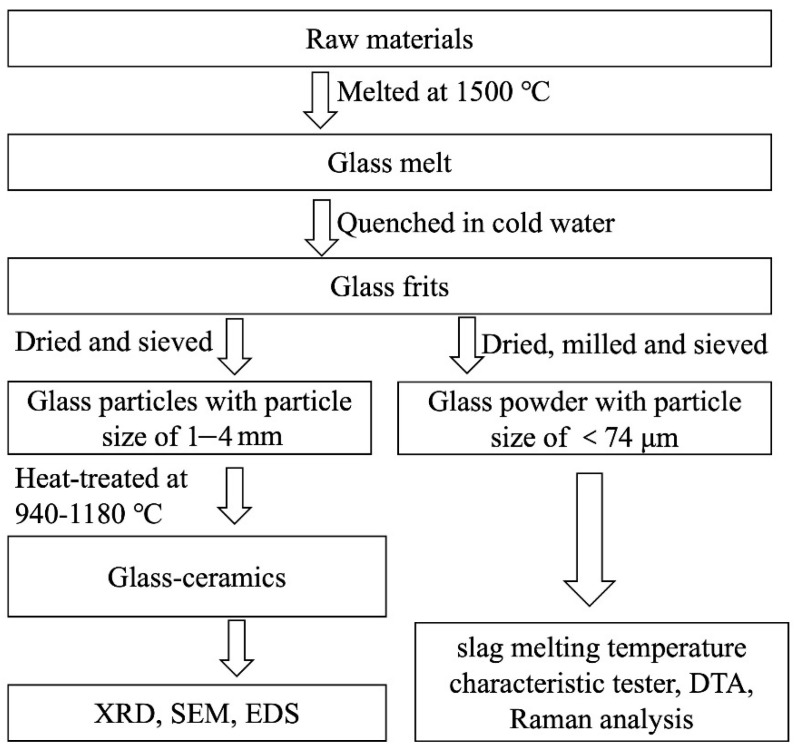
The process flow diagram of the experiment.

**Figure 2 materials-14-00681-f002:**
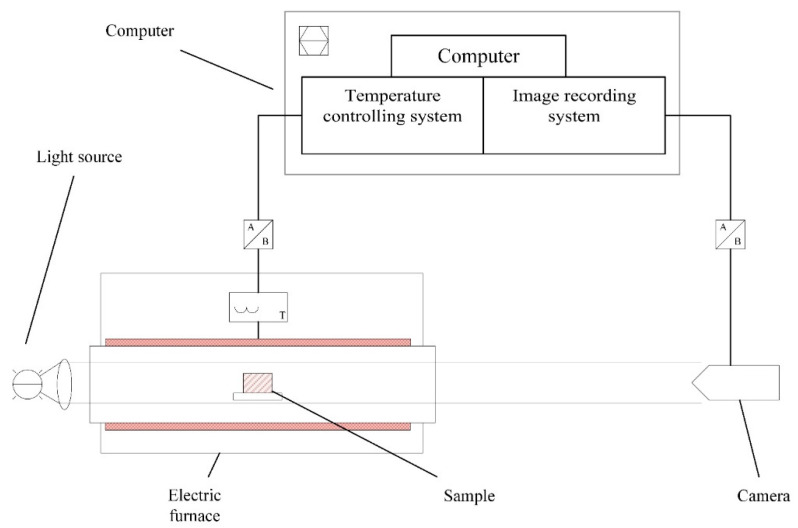
Schematic representation of a slag melting temperature characteristic tester for sintering analysis.

**Figure 3 materials-14-00681-f003:**
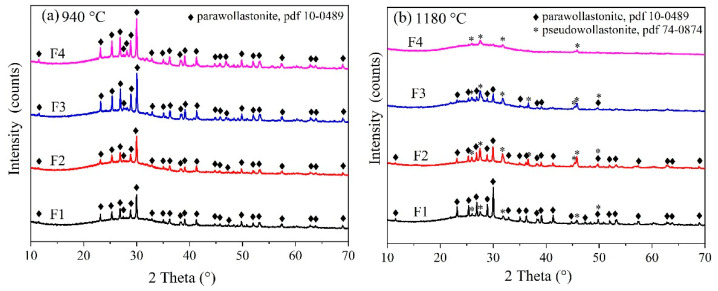
XRD results of wollastonite glass-ceramics obtained by heat treatment at different temperatures (**a**) 940 °C, (**b**) 1180 °C.

**Figure 4 materials-14-00681-f004:**
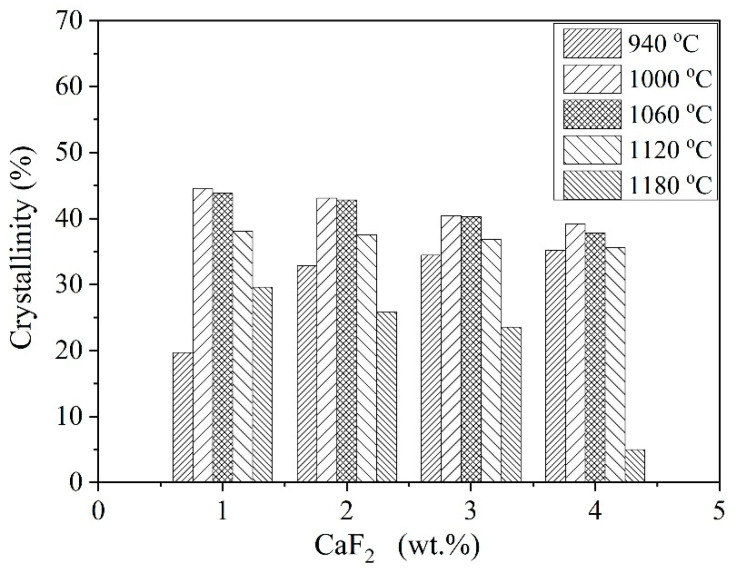
The crystallinity of wollastonite glass-ceramics calculated by Rietveld refinement from the XRD patterns.

**Figure 5 materials-14-00681-f005:**
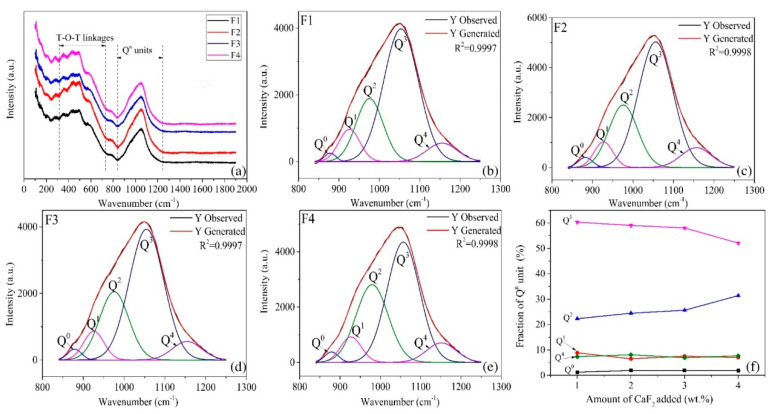
(**a**) Raman spectra of the quenched glass melts F1–F4, (**b**–**e**) deconvolution of the Raman spectra for the glass melts, (**f**) relative fraction of the Q^n^ units for the glass melts as a function of the CaF_2_ content.

**Figure 6 materials-14-00681-f006:**
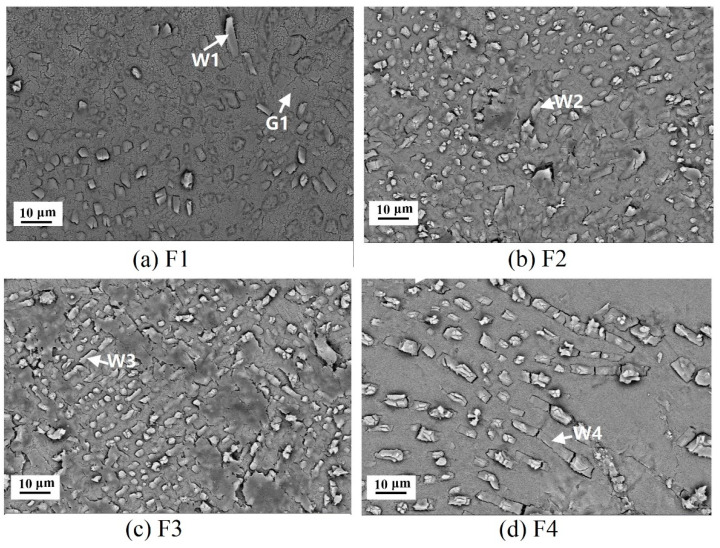
The microstructure images of wollastonite glass-ceramics with different CaF_2_ addition obtained by heat treatment at 1120 °C. (**a**): sample F1; (**b**): sample F2; (**c**): sample F3; (**d**): sample F4.

**Figure 7 materials-14-00681-f007:**
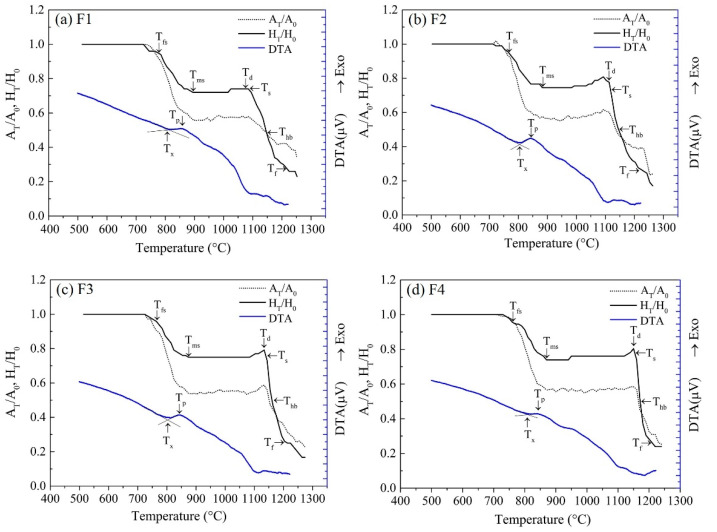
Shrinkage and DTA curves of samples F1–F4 (Note: T_x_—onset temperature of crystallization; T_p_—crystallization peak temperature; T_fs_—onset temperature of sintering; T_ms_—temperature of maximum shrinkage before softening; T_d_—onset temperature of softening; T_s_—softening temperature; T_hb_—hemispheric temperature; T_f_—flowing temperature; A_T_, H_T_ are the area and height of sample images at temperature T, respectively). (**a**) sample F1; (**b**) sample F2; (**c**) sample F3; (**d**) sample F4.

**Figure 8 materials-14-00681-f008:**
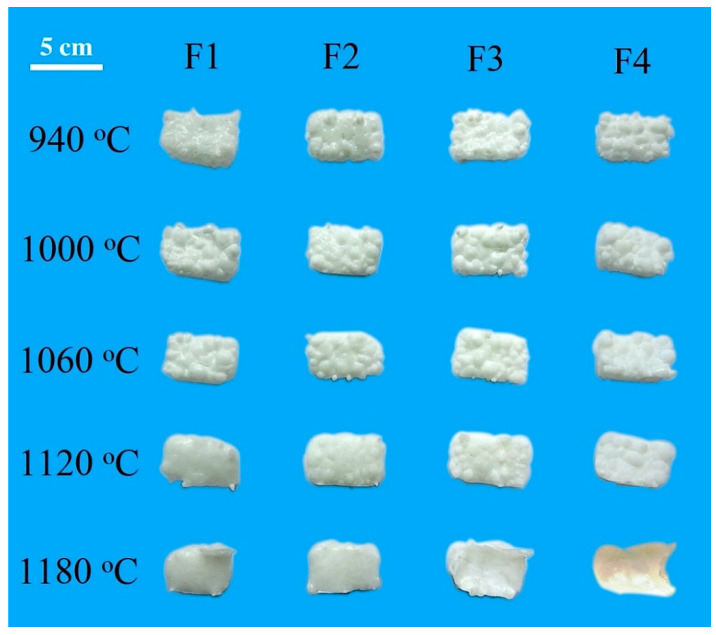
The influence of CaF_2_ addition on the appearance of the glass-ceramic samples at different temperatures.

**Table 1 materials-14-00681-t001:** Chemical compositions of the designed parent glasses.

Sample No.	Mass (GRAM)									
	100 g									CaF_2_
	Al_2_O_3_	SiO_2_	CaO	K_2_O	Na_2_O	BaO	B_2_O_3_	ZnO	Sb_2_O_3_	
F1	7.04	58.84	17.03	2.10	2.94	4.10	1.01	6.50	0.44	1
F2	7.04	58.84	17.03	2.10	2.94	4.10	1.01	6.50	0.44	2
F3	7.04	58.84	17.03	2.10	2.94	4.10	1.01	6.50	0.44	3
F4	7.04	58.84	17.03	2.10	2.94	4.10	1.01	6.50	0.44	4

**Table 2 materials-14-00681-t002:** The interpretations of the characteristic temperatures correlated to sintering events.

Characteristic Temperatures	Interpretations
500 °C	initial height H_0_, initial area S_0_
The onset temperature of sintering, T_fs_	where the height of sample H_fs_ becomes 95–97% of H_0_
The temperature of maximum shrinkage, T_ms_	the sample experiences the maximum shrinkage before softening, where its height stops decreasing and becomes constant
The onset temperature of softening, T_d_	where the sample begins to soften and the edges of the sample becomes rounded [14]
The softening temperature, T_s_	where the height of sample H_s_ becomes 75% of H_0_
The hemispheric temperature, T_hb_	where the height of sample H_hb_ becomes 50% of H_0_
The flowing temperature, T_f_	where the height of sample H_f_ becomes 25% of H_0_

**Table 3 materials-14-00681-t003:** Semiquantitative energy spectrum analysis of the selected phases shown in Figure 6.

Spot No.	Phase	Element (wt.%)								
		O	Si	Ca	Na	Al	F	Zn	Ba	K
G1	Glass matrix	25.53	21.01	5.87	1.43	4.71	27.31	7.37	5.35	1.43
W1	wollastonite	6.81	4.56	24.50	0.78	0.73	62.61	-	-	-
W2		11.49	4.73	17.62	1.29	0.84	62.75	-	-	-
W3		10.58	4.58	22.16	1.36	1.13	60.20	-	-	-
W4		6.74	5.65	23.41	1.09	0.81	62.29	-	-	-

**Table 4 materials-14-00681-t004:** Characteristic temperatures for samples F1–F4.

Items	Sample No.			
	F1	F2	F3	F4
The onset temperature of crystallization, T_x_/°C	815	808	804	805
Crystallization peak temperature, T_p_/°C	858	841	840	840
The onset temperature of sintering, T_fs_/°C	775	765	764	760
The temperature of maximum shrinkage, T_ms_/°C	895	883	874	870
The ability of crystallization vs. sintering, ΔT_1_= T_ms_ − T_x_/°C	37	42	34	30
The onset temperature of softening, T_d_/°C	1075	1113	1134	1150
The softening temperature, T_s_/°C	1085	1118	1139	1155
The hemispheric temperature, T_hb_/°C	1145	1148	1159	1170
The flowing temperature, T_f_/°C	1215	1213	1204	1200
Dense sintering temperature range, ΔT_2_= T_hb_ − T_d_/°C	70	35	25	20

## Data Availability

The data presented in this study are available in this article.

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
