# Peer review of "The Concurrent Sintering-Crystallization Behavior of Fluoride-Containing Wollastonite Glass-Ceramics"

_materials, 2021, doi:10.3390/ma14030681_

Round 1

Reviewer 1 Report

Dear Sir,
The paper is interesting and well prepared however, it can be improved in some areas as follows:

The title

The number of words in the title should be minimized.

 In the abstract

Abstract should be in Present tense while conclusions are given in the Past tenses.

keywords should be arranged alphabetically.

Introduction section:

Some minor grammatical should be corrected in the manuscript.

  • More general explanation regarding to glass ceramic materials can be useful to the reader combined with their applications such implementation as electronic devices, biomedical applications, sensors etc. some advised reference such as:  https://doi.org/10.3390/s17112538

In the other sections

  • Schematic diagram for the experimental part could be useful for the reader
  • State the model, the company and country of origin of any instrument used in the research such as muffle furnace.
  • Could the author give more information about the DTA analysis?
  • 1 is not in a good quality and cannot be distinguished if it printed in black and white.
  • Please specify how many replicates did you perform and provide some information about the measurement error we can encounter.
  • I recommend that in the characterization part it can be divided into subsections such as XRD analysis, SEM etc.
  • Future work should be mentioned at the end of the conclusion

References

Kindly FORMAT the references correctly according to the author’s guide.

Regards

Reviewer 2 Report

Unfortunately, the work “Revealing the effect of CaF2 on the concurrent sintering-crystallization competitive behaviour of wollastonite glass-ceramics” by Chuanhui Li, Peng Li, Jianliang Zhang, Fengjuan Pei, Wei Zhao, Bingji Yan, Hongwei Guo is not appropriated for publication.

The study is related to the well-known building glass-ceramics “Neoparies” of NEG, which is produced in Japan from half century. Similar products are fabricated also in China from decades. Alternative compositions, based on various industrial wastes, are also studied and reported.

In the present work are shown some laboratory result for the obvious effect of CaF2 addition on the sintering and crystallization. Some of experiments are made using fine powders, which give no useful information for the production of similar materials, which are manufactured by sinter-crystallization of glass grains (1-7 mm) placed loosely in refractory forms (up to 2 m2).

The crystallization experiments also don’t give important information, because at lower temperature the crystallization time is not sufficient to complete the crystallization. Moreover, if preliminary crystallization step at lower temperature is use it might hinder the subsequent sintering at higher temperatures.

Since the marble-like effect of Neoparies is due to surface crystallization, the bulk crystallization as well as the liquid-liquid immiscibility, created by addition of CaF2, can be avoided!

No results for the degree of sintering are reported.

The initial glass (i.e. without CaF2) is not studied.

A part of used terminology is not correct.

Reviewer 3 Report

The reviewed paper tries to explain the influence of CaF2 on the  sintering-crystallization competitive behavior of wollastonite glass-ceramics. In my opinion, the paper must be reworked before it is published.

The authors propose the mechanism of replacing oxygen ions with fluorine ones, however, the discussion lacks any evidence that such a mechanism occurs in this case study. In my opinion, additional research methods, e.g. IR/Raman spectroscopy and/or NMR spectroscopy, should be used for its verification.

Another thing is the increase / decrease of the crystallinity of the samples. How was it calculated? This also should be explained.

Reviewer 4 Report

The paper presents interesting data relevant to Materials. However, the manuscript needs a thorough revision before accepted to publication.

Language is sometimes weird and confusing and needs therefore editorial work. However, the most striking parts which need attention are as follows:  

Table 1: Mass fractions are wrongly given, and the sum is everywhere larger than 100%.

The samples obtained were powders with sizes of 1 to 4 mm as it follows from lines 63-64, then how these samples were fractured and polished?

Lines 68 and 74: please explain how were obtained powders with sizes < 74um (the um shall be corrected to μm most probably) and ≤ 74 μm.

Line 76 onward: there are no DTA data presented.

Line 79: the claim “heat signal for the glass transition” is wrong as there is not any latent heat of vitrification.

The content of Table 2 is fully unclear and confusing. There is not any explanation of symbols such as H with subscripts.

Line 88: Explain how was obtained the glass powder (particle size≤0.074).

Line 96 and Fig. 1 – the XRD peaks are not identified. These shall be shown explicit in the figure.

Lines 97-98: there are very many peaks in the XRD patterns against of the claim.

Lines 119-120: Provide and explain the equation used to calculate the crystallinity shown in Fig. 2.

Line 133: The claim “decrease melting point, with a consequent lowering of viscosity” is wrongly turning the case with the consequence. This is because the bond-breaking process (herewith caused by CaF2) is lowering the viscosity, see for example the overview on viscosity: Viscous flow and the viscosity of melts and glasses. Physics and Chemistry of Glasses, 53 (4) 143-150 (2012).

Lines 151-152: The claim “Since diffusion is responsible for sintering” is incomplete because both diffusion and viscous flow are promoting sintering not the diffusion alone.

Line 153: Authors are noting that “CaF2 in glass acts as a powerful network disrupter” and namely this leads to a decrease of the viscosity – see the Reference Physics and Chemistry of Glasses, 53 (4) 143-150 (2012).

Figure 4: The Tg’s are not specified, the notations “AT/A0-relative area; HT/H0-relative height” are also not explained.

Figure 5: Sizes are not shown.

Reviewer 5 Report

I recommend making the following corrections:

  1. Line 19: smooth and flat surface
  2. Line 33: wollastonite [6], whereas the
  3. Line 89: was compressed into a cylinder
  4. Line 96: Figure 1 shows the XRD results (Fig 3 shows the microstructure)
  5. Line 155: between adjacent particles were developed
  6. Line 160: Based on the DTA curves
  7. Lines 180-181: It might be the potential reason for the samples
  8. Line 208: had a broad temperature
  9. Line 220-222: Whereas F3 and F4 samples, owing to the lower flowing temperature, had a low crystallinity and exhibited over-burnt surface

Figure 2 is mentioned nowhere in the text.

In lines 188-189, it is written: „The latter is due to the degraded refractoriness of glass-ceramics when CaF2 is added.” meaning the flowing temperature is decreased due to the degraded refractoriness. It should be explained in more detail how the refractoriness could influence the melting behaviour of glass-ceramics.

It is not clear from the context, that „Generally, the dense sintering temperature ΔT2 of glass-ceramics should be greater than 30 °C.” (Line 196) is a conclusion from Table 4, or is a piece of information from the literature review?

In lines 205-206, the Authors write: 2However, as shown in Figure 5, the rough surface of the glass-ceramic samples was observed.” – It is not clear, which samples have a rough surface. The quality of Figure 5 needs to be improved and a scale bar must be placed. 

Round 2

Reviewer 1 Report

Dear Sir,

I would like to thanks the authors for the modification I think the paper can be suitable for publication in its latest version.

Regards

Author Response

The reviewer’s work is greatly appreciated.

Reviewer 3 Report

I accept the answer to the second point, but still insist that additional studies should be carried out in order to determine substitution process.

In addition if, like authors suggest, the studied materials are very similar, it should be fairly easy to provide such results.

Reviewer 4 Report

Accept in the revised form

Author Response

(The authors gave the same response as above.)

Round 3

Reviewer 3 Report

Thank you for adding Raman results. Now, in my opinion, manuscript is ready for publishing.